# Prenatal and Peripartum Exposure to Antibiotics and Cesarean Section Delivery Are Associated with Differences in Diversity and Composition of the Infant Meconium Microbiome

**DOI:** 10.3390/microorganisms8020179

**Published:** 2020-01-27

**Authors:** Wendy S.W. Wong, Priya Sabu, Varsha Deopujari, Shira Levy, Ankit A. Shah, Nicole Clemency, Marina Provenzano, Reem Saadoon, Akhil Munagala, Robin Baker, Rajiv Baveja, Noel T. Mueller, Maria Gloria Dominguez-Bello, Kathi Huddleston, John E. Niederhuber, Suchitra K. Hourigan

**Affiliations:** 1Inova Translational Medicine Institute, Falls Church, Inova Fairfax Hospital, Falls Church, VA 22042, USAVarsha.Deopujari@inova.org (V.D.); shira.levy@inova.org (S.L.); nclemency@gmail.com (N.C.); Marina.Provenzano@inova.org (M.P.); reem.saadoon91@gmail.com (R.S.); muakhile@umich.edu (A.M.); kathihuddleston@cox.net (K.H.); jniederhuber@msn.com (J.E.N.); 2School of Medicine, Virginia Commonwealth University, Richmond, VA 23298, USA; sabup@mymail.vcu.edu; 3Division of Maternal Fetal Medicine, Department of Ob/Gyn, Inova Fairfax Hospital, Falls Church, VA 22042, USA; shah@mypanova.com; 4Fairfax Neonatal Associates, Falls Church, VA 22042, USA; RBaker@fnapc.com (R.B.); RBaveja@fnapc.com (R.B.); 5Department of Epidemiology, Johns Hopkins School of Public Health, Baltimore, MD 21205, USA; noeltmueller@jhu.edu; 6Department of Biochemistry and Microbiology, Rutgers University, New Brunswick, NJ 08901, USA; mg.dominguez-bello@rutgers.edu; 7College of Health and Human Services, George Mason University, Fairfax, VA 22030, USA; 8Public Health Sciences, School of Medicine, University of Virginia, Charlottesville, VA 22908, USA; 9Johns Hopkins School of Medicine, Baltimore, MD 21205, USA; 10Inova Children’s Hospital, Inova Fairfax Hospital, Falls Church, VA 22042, USA; 11Pediatric Specialists of Virginia, Fairfax, VA 22031, USA

**Keywords:** microbiome, neonate, infant, pediatrics, antibiotics, delivery mode

## Abstract

The meconium microbiome may provide insight into intrauterine and peripartum exposures and the very earliest intestinal pioneering microbes. Prenatal antibiotics have been associated with later obesity in children, which is thought to be driven by microbiome dependent mechanisms. However, there is little data regarding associations of prenatal or peripartum antibiotic exposure, with or without cesarean section (CS), with the features of the meconium microbiome. In this study, 16S ribosomal RNA gene sequencing was performed on bacterial DNA of meconium samples from 105 infants in a birth cohort study. After multivariable adjustment, delivery mode (*p* = 0.044), prenatal antibiotic use (*p* = 0.005) and peripartum antibiotic use (*p* < 0.001) were associated with beta diversity of the infant meconium microbiome. CS (vs. vaginal delivery) and peripartum antibiotics were also associated with greater alpha diversity of the meconium microbiome (Shannon and Simpson, *p* < 0.05). Meconium from infants born by CS (vs. vaginal delivery) had lower relative abundance of the genus *Escherichia* (*p* < 0.001). Prenatal antibiotic use and peripartum antibiotic use (both in the overall analytic sample and when restricting to vaginally delivered infants) were associated with differential abundance of several bacterial taxa in the meconium. Bacterial taxa in the meconium microbiome were also differentially associated with infant excess weight at 12 months of age, however, sample size was limited for this comparison. In conclusion, prenatal and peripartum antibiotic use along with CS delivery were associated with differences in the diversity and composition of the meconium microbiome. Whether or not these differences in the meconium microbiome portend risk for long-term health outcomes warrants further exploration.

## 1. Introduction

The human gut microbiome undergoes rapid dynamic development in the early years of life, reaching a stable state with a diversity and complexity resembling adult gut microbiota generally by the age of three [1]. These dynamic changes correspond with the various exposures in early life, including the delivery mode (vaginal delivery (VD) versus caesarean section (CS)) and dietary changes (breast feeding versus formula feeding and timing of introduction of solid food) [2,3,4]. Indeed, it is thought that the critical period of initial microbiome development is influenced by the priority effects, in which the order and timing of the species arrival determines how species affect one another, with disruption to this order having long-lasting consequences [5]. While postnatal events are thought to have the greatest influence, recent evidence has emerged to suggest that prenatal factors may also play a role in infant microbiome development [6]. Analysis of the meconium (the first stool passed by an infant) microbiome may help give insight into the very early development of the gut microbiome by pioneering microbes in relation to intrauterine events. 

Aberrant early intestinal microbiome development has been associated with specific childhood diseases, such as obesity. Childhood obesity has become a critical public health problem [7,8], with evidence that very early infant and childhood obesity is predictive of obesity in later in life [9,10,11]. A critical window in very early life is hypothesized to be especially predictive of later obesity risk due to the establishment of the gut microbiome with associated metabolic programming and the epigenetic changes that can occur during the perinatal and neonatal period [1,11,12]. Analysis of the meconium microbiome may give further understanding into the very early microbiome development that may contribute to the development of obesity and other microbiome-associated diseases.

Early life antibiotic use, including during the prenatal and peripartum period, has been associated with higher obesity risk in studies [13,14,15,16,17]. Likewise, CS has also been associated with higher risk of obesity [17,18,19,20], but it is unclear how much of this association is explained by peripartum antibiotics given during delivery. Antibiotics can have a profound effect on the intestinal microbiome, with both short and long-term changes seen in adults, children, and infants [21,22,23]. It is hypothesized that early antibiotic exposure causes detrimental gut microbiota perturbations that ‘program’ the host to an obesity-prone metabolic phenotype, which persists after the discontinuation of antibiotics. Murine models support this hypothesis, suggesting a causal role for antibiotic modified microbiota in the development of excess weight [24,25]. However, there are a lack of human studies investigating the effect of prenatal and peripartum antibiotic exposure on the early microbiome. Studying the meconium microbiome offers a window into how these intrauterine exposures may affect the seeding of the earliest gut microbiome.

The study reported in this manuscript was undertaken to investigate the association of prenatal and peripartum antibiotic exposure and delivery mode on the meconium microbiome, and to determine if the meconium microbiome was associated with excess weight of the infant at 1 year of life.

## 2. Methods

### 2.1. Subjects and Sample Collection 

Mothers were recruited prenatally in an ongoing longitudinal childhood health cohort study within the Inova Health System “The First 1000 Days of Life and Beyond”. Informed consent was obtained from all subjects. All methods were carried out in accordance with relevant guidelines and regulations. All experimental protocols were approved by the Inova Health System Institutional Review Board (approval #15-1804). Infants had meconium collected (defined as the first stool passed, other than meconium staining of amniotic fluid). Meconium was stored in Eppendorf Tubes at −80°C within 12 hours of collection. Only full term infants (gestational age ≥37 weeks) were included.

Demographic information was collected including infant sex, maternal ethnicity, and gestational age at delivery by questionnaire and the electronic medical record. Detailed information was recorded regarding delivery (VD, scheduled CS (scheduled in advance, no rupture of membranes) and emergency CS (not planned, due to dystocia or for mother/fetal distress, before or after the rupture of membranes). Pregnancy details collected from the electronic medical records included the use of prenatal antibiotics (defined as antibiotics given to mother from conception to 2 days prior to delivery) and peripartum antibiotics (defined as antibiotics given to the mother from 2 days prior to delivery to during the delivery). Weight gain during pregnancy was calculated with pre-pregnancy weight and admission weight at delivery, noted in electronic health records [26]. The Health and Medicine Division of the National Academies weight gain recommendations were used to determine whether mothers met the recommended weight gain for their pre-pregnancy body mass index (BMI). Gestational weight gain recommendations were categorized into: “lower than recommended range”, “within recommended range,” and “greater than recommended range” [27]. Parentally reported anthropometrics for infants at 12 months of age were recorded, outliers were removed for reported lengths using the interquartile range, and weight for length percentiles (WFLP) at 12 months were calculated with the World Health Organization sex-specific growth charts, as previously validated in this cohort [28]. In this study, excess weight was defined as WFLP ≥ 95th% and risk for excess weight as WFLP ≥ 85th%.

### 2.2. DNA Extraction

Meconium samples stored at −80 °C were thawed on ice and suspended in ASL buffer (Qiagen, Valencia, CA, USA) at a ratio of 2.5 mL ASL to 0.5 g of meconium [29]. Tube-stored samples were aliquoted into 2 mL Matrix A lysis tubes (MP Biomedical, Santa Ana, CA, USA). Samples were homogenized for 10 min on an oscillating vortex at maximum speed and placed briefly in a flash spinner. Twenty-five microliters of lysozyme at 20 mg/mL was added to the tubes and inverted 10 times to mix. Samples were placed on a shaking heat block at 95 °C for 5 min at 2000 rpm, cooled on ice for 2–5 min, and centrifuged at 20,000× *g* for 2 min. Supernatant was removed, avoiding the pellet, and placed in a new ET with one Inhibit X tablet (Qiagen). Samples were vortexed until the tablet was dissolved and incubated at room temperature for 3 min. Samples were centrifuged at 20,000× *g* for 2 min; supernatant was removed and put into a new Eppendorf Tube and centrifuged again at 20,000× *g* for 3 min. The Qiagen QIAmp DNA Stool Mini kit protocol then followed (Qiagen). 

### 2.3. 16S rRNA Gene Sequencing

Sequencing libraries were prepared using a Nextera XT kit (Illumina, San Diego, CA, USA) according to the 16S Metagenomic Illumina Library Preparation Protocol for sequencing the variable V3 and V4 regions of the 16S rRNA gene. If samples failed QC, library preparation was completed again with the “PCR 1” using Hemo KlenTaq® (New England Biolabs, Ipswich, MA, USA) for the PCR reaction. Hemo KlenTaq® is known to work well with sample containing PCR inhibitors, especially bilirubin, which is highly present in meconium samples. Each specimen/condition passing QC was then sequenced on the Illumina MiSeq platform (Illumina) with paired-end reads 301 bp (600 cycles). Only 3 meconium samples failed sequencing, leaving a total of 105 for analysis. Sequencing of negative controls of lysis buffer undergoing the above DNA extraction process was performed. Positive control samples of *Staphylococcus aureus* and *Escherichia coli* were also included.

### 2.4. 16S Data Processing and Analysis

Demuliplexed reads from each sample were treated as single-ended reads and bacterial operational taxonomic units (OTUs) were picked using the open reference method using QIIME 1.9 [30] with Greengenes 16S rRNA database [31]. OTUs that were not seen more than 3 times in at least 20% of the samples were removed. 

The OTU counts table and the phylogenetic tree created by QIIME were imported into R bioconductor package phyloseq (Version 1.25.2) for data analysis. All the figures in the manuscript were generated by R Version 3.4.0. A single rarefaction at 7324 was performed (Appendix A) to normalize the counts. 

Beta diversity was measured by the unweighted unifrac distance in order to capture the differences in the low abundance features. PERMANOVA was performed using the adonis function in the R package Vegan 2.5-3 to compare the beta distances between multiple clinical factors. The adonis function in R package Vegan 2.5-3 was also used to perform permutational multivariate analysis of variance using the unweighted unifrac distance matrices. Alpha diversity was measured by the number of OTUs, and the Fisher, Shannon, and Simpson indices. Two sample Wilcoxon test was used to access differences in alpha diversity between groups. 

Normalized counts for each sample were used to calculate the abundance of bacteria at the phylum, class, order, family, and genus levels. A two sample Wilcoxon test was used to compare the differences in relative abundance between groups at the phylum and genus levels using the normalized counts. Raw read counts for each OTU for each sample were analyzed by R Bioconductor package EdgeR 3.20.9 to discover differentially abundant OTUs according to clinical factors of interest. Benjamini and Hochberg adjustment was performed for multiple testings [32]. 

The 16S rRNA sequence data are available in NCBI Sequence Read Archive (SRA), PRJNA600283.

## 3. Results

### 3.1. Subject Demographics and Clinical Characteristics

Meconium was collected within 48 hours of life and was sequenced successfully from 105 infants. Forty-five (43%) were male; 62 (59%) were born by VD and 43 (41%) by CS. Of those delivered by CS, 27/43 (63%) were by a scheduled CS and 16/43 (37%) by emergency CS. Table 1 shows the sample distributions of clinical factors of interest between CS and VD. Mothers of infants born by CS received significantly more peripartum antibiotics than those born by VD (43/43 (100%) in CS vs. 19/62 (31%) in VD, *p* < 0.001). Of those receiving peripartum antibiotics for VD, all were for detected Group B *Streptococcus* (GBS) prophylaxis. Women without a penicillin allergy undergoing a scheduled CS received a first generation cephalosporin within 60 minutes prior to the start of CS (cefazolin 1 g IV for women <80 kg, 2 g for women ≥80 kg, and 3 g for women ≥120 kg). Those who were penicillin allergic (2/27) received clindamycin (900 mg intravenously) and gentamicin (5 mg/kg intravenously). Women undergoing emergency CS also received a first generation cephalosporin; additional azithromycin was given in 12/16 mothers (500 mg intravenously). All 19 women who had VD and received antibiotics for GBS prophylaxis received penicillin (Penicillin G 5 million units intravenously initial dose, then 2.5 to 3 million units intravenously every four hours until delivery). Dosing for peripartum antibiotics was confirmed in the medical administration record, as these were inpatient medications. Seventeen out of 105 (16%) of mothers received prenatal antibiotics; these were given for urinary tract infections (10/17) and respiratory infections (7/17). Penicillins, cephalosporins, or macrolides were used in all 17 cases. One out of 17 received antibiotics in the 1st trimester, 7/17 in 2nd trimester, 8/17 in 3rd trimester, and 1/17 in both the 1st and 2nd trimester. While these prenatal antibiotics were documented in the electronic medical record, dosing and duration could often not be confirmed, as they were not part of the medical administration record due to being outpatient medications.

### 3.2. Beta Diversity

In the 105 meconium samples, each of the clinical variables was examined in relation to beta diversity using the Adonis test with 9999 permutations. Beta diversity was significantly associated with delivery mode (CS versus VD), prenatal antibiotic use, and peripartum antibiotic use (all *p* < 0.05). No significant differences were found in the sex, maternal pre-pregnancy BMI, maternal weight gain during pregnancy, weight status of the infant at 12 months of age, and scheduled CS versus emergency CS (all uncorrected p > 0.05). In multivariable analyses adjusted for covariates that were significant in univariable models, delivery mode (*p* = 0.044), prenatal antibiotic use (*p* = 0.005), and peripartum antibiotic use (*p* < 0.001) remained significant.

### 3.3. Alpha Diversity

When analyzing the alpha diversity of meconium, infants delivered born by CS had higher Shannon (*p* = 0.013) and Simpson (*p* = 0.002) indices compared to VD, but no difference in the number of observed OTUs (*p* = 0.55) or Fisher (*p* = 0.58) diversity. Use of peripartum antibiotics was associated with higher Shannon and Simpson diversity, but a lower number of observed OTUs and Fisher diversity (Figure 1a). While all four alpha diversity measures indicate the OTU microbial richness of the samples, the Shannon diversity and Simpson diversity also measures the OTU microbial evenness of a sample. Further, the Simpson index gives more weight to the more abundant OTUs. Peripartum antibiotic use was associated with increased Shannon and Simpson diversity, but not Fisher or observed OTUs, suggesting that peripartum antibiotics may be associated with a decrease of some of the rarer taxa. While not meeting statistical significance, prenatal antibiotic use was associated with a trend toward lower Shannon and Simpson diversities (*p* = 0.07 and *p* = 0.09, respectively). Shannon diversity was also higher in samples from non-Hispanic mothers (*p* = 0.04), but sample size for ethnicity analysis was limited. No significant differences in any of the alpha diversity measures were observed based on sex, maternal weight gain during pregnancy, and weight status of the infant at 12 months of age.

Next, whether these alpha diversity associations remained significant when considering only infants born by VD was investigated (Figure 1b). Prenatal antibiotic use, weight status of the infant at 12 months, and maternal weight gain during pregnancy was not associated with alpha diversity among VD infants. The use of peripartum antibiotics was associated with higher alpha diversity according to all four measures (*p* < 0.05) among VD infants. Observed OTUs, and Shannon and Fisher indices were also higher in samples from non-Hispanic mothers (*p* < 0.05), however, the peripartum antibiotic use in infants born by VD was highly skewed between the Hispanic and non-Hispanic populations, with 9 out of 20 (45%) non-Hispanic mothers using peripartum antibiotics compared with only 2 out of 13 (15%) Hispanic mothers. Therefore, whether there was a difference in alpha diversity in the samples from mothers that did not use peripartum antibiotics was further examined (11 Hispanic mothers and 11 non-Hispanic mothers). Two sample t-tests demonstrated that, while the average alpha diversity in the four alpha diversity measures were still higher in samples from infants born from non-Hispanic mothers, the differences were no longer significant. It was therefore concluded that the difference observed between ethnicity in the mothers may be driven by the peripartum antibiotic use.

### 3.4. Relative Abundance of Taxa

Overall, in full term meconium samples, Proteobacteria had the highest mean abundance at the phylum level (65.2%), followed by Firmicutes (26.3%), and Bacteroidetes (4.2%). At the genus level, *Escherichia* had the highest mean abundance (28.5%), followed by *Methylobacterium* (15.7%) and *Streptococcus* (7.2%).

A difference was not observed in relative abundance of any bacterial phylum in the meconium of VD vs. CS infants (p > 0.05, two sample Wilcoxon test). At the genus level, the relative abundance of *Escherichia* and *Bacteroides* were higher in VD (FDR adjusted *p* < 0.05). *Escherichia* in particular was markedly higher in VD (38%) vs. CS (15%) infants (FDR adjusted *p* < 0.001), Figure 2). The relative abundance of *Methylobacterium* and *Veillonella* were higher in CS meconium samples (adjusted *p* < 0.05), with *Methylobacterium* being markedly higher in CS compared to VD (27% CS vs. 8% VD, FDR adjusted *p* = 0.001; Figure 3).

There were 205 OTUs that had a significantly different relative abundance between the two delivery modes (FDR < 0.05). Among the top 10 most significant OTUs, 8 were from the genus *Streptococcus* (Figure 4), which was enriched in CS; the other 2 were *Agrobacterium,* which was enriched in samples from CS infants, and *Bacteroides*, which was enriched in samples from VD infants (Figure 4). Significantly different relative abundances of OTUs between emergent CS and scheduled CS were then examined. The top 10 OTUs, with their genus and Phylum, are shown in Appendix A. OTUs from *Streptococcus, Staphylococcus, Veillonella, Bacteroides*, *Enterococcus,* and *Corynebacterium* were higher in scheduled CS; OTUs from *Propionibacterium, Stenotrophomonas, Acinetobacter* and *Methylobacterium* were higher in emergency CS.

Next, antibiotic use was examined. A total of 126 bacterial OTUs were differentially abundant between meconium samples from infants delivered to mothers with vs. without prenatal antibiotic usage. The number of OTUs decreased to 75 if only infants born by VD were considered (Appendix A). The top three most differentiated OTUs between infants born to mothers who used vs. did not use prenatal antibiotics were the same if infants born by CS were excluded or not; an OTU from the genus *Clostridium* and 2 OTUs from the family Enterobacteriaceae with unspecified genus were enriched in meconium samples from infants born to mothers who used prenatal antibiotics. 

A total of 214 OTUs were significantly different in relative abundance between samples from mothers with vs. without peripartum antibiotics use; the number of significant OTUs decreased to 53 for infants born by VD only (Appendix A). The top three most differentiated OTUs between peripartum antibiotic use groups were the same if infants born by CS were excluded or not. The OTU belonging to the genus *Enterococcus* was lowes in the meconium of infants born to mothers that used peripartum antibiotics, whereas the OTUs belonging to *Agrobacterium* and *Streptococcus* were higher.

Differentially abundant OTUs were also examined between other clinically relevant factors. Appendix A show that OTUs were differentially abundant by infant sex, infant weight status at 12 months of age, and maternal weight gain during pregnancy, respectively. Infants with excess weight at 12 months had a higher number of OTUs from the genera *Streptococcus, Klebsiella, Escherichia, Veillonella,* and *Acinetobacter*. There were more OTUs from the genera *Pseudomonas* and *Enterococcus* and fewer OTUs from *Escherichia* and *Propionibacterium* in the meconium of infants born to mothers with excess weight gain. 

## 4. Discussion

In our birth cohort of full-term mother-child dyads, it was found that prenatal antibiotic use, peripartum antibiotic use, and delivery mode were associated with features of the meconium microbiome. The beta diversity of meconium, after multivariable adjustment, was associated with delivery mode, with prenatal antibiotic use and with peripartum antibiotic use. CS and peripartum antibiotics were associated with greater alpha diversity (Shannon and Simpson), and CS samples had a lower relative abundance of the genus *Escherichia*. Prior studies support the notion that the microbiome present in the first passage of newborn meconium may reflect the infant’s intrauterine and perinatal environmental exposures [33]. This study adds to previously reported studies and suggests that prenatal and peripartum maternal antibiotics may reach the fetus and, in addition to the mode of delivery, alter the structure and composition of the meconium microbiome. 

Peripartum antibiotics are given to ≥40% of pregnant women in the USA and have successfully decreased the rate of neonatal GBS infection and maternal post CS infection [34,35,36]. It is thought that peripartum antibiotics affect the bacterial populations of the mother’s vaginal canal, gut, and skin, and these altered bacteria are transmitted to the baby during and after delivery [37]. This hypothesis is supported by a study from from Tanaka et al. [38], in which infants exposed to antibiotics had aberrant early gut microbiome development over the first two months postnatally. However, our study suggests that a brief exposure to antibiotics before birth may reach the fetus leading to the changes in the meconium microbiome.

In examining the association of peripartum antibiotics with the meconium microbiome, it is important to account for delivery mode, which can result in a lack of exposure to maternal vaginal and perineal microbes [2]. Indeed, in murine models, CS without antibiotic exposure impacted the development of the gut microbiota and was associated with increased body weight [39]. Our results differ from Mueller et al. who only found significant differences in the microbiome by delivery mode in transitional stool and not meconium [40], but concurs with differences seen in the microbiome by delivery mode when shotgun metagenomic sequencing was used [41]. Differences may be due to the timing of meconium collection across studies. In our study, lower levels of *Escherichia* were seen in the meconium from infants born by CS. This is a shift from the expected “normal”, very early colonization pattern where there is a predominance of facultative anaerobes, such as species from *Escherichia*, as seen in infants born by VD [42]. 

Prenatal antibiotic exposure was also associated with changes in the meconium microbiome, although to a lesser extent, with differences in beta diversity and relative abundance of certain taxa and a trend towards lower alpha diversity. Prenatal antibiotic exposure has been associated with later obesity in children [17], however this affect may be mediated more through immunologic and/or metabolic programming occurring before birth than on pioneering intestinal microbes. Indeed, aberrant methylation at imprinted genes has previously been associated with prenatal antibiotic use [43]. Furthermore, in murine models immune and inflammatory responses in infants were dependent on prenatal maternal microbiota [44], which can be affected by prenatal antibiotic exposure.

Prenatal and peripartum antibiotics, in addition to CS delivery, have been associated with obesity [13,14,15,16,17,18,19,20], with murine models suggesting a causal role for antibiotic altered microbiota [24,25]. While changes in the meconium microbiome were clearly detected with prenatal and peripartum antibiotic use, no differences were seen in relation to the infant weight status at 12 months of age in alpha and beta diversity, and only subtle changes were seen in the relative abundance of taxa between normal weight and overweight/obese infants. This suggests that antibiotic exposure may program the host to an obesity related phenotype, which is reflected in the subsequent postnatal microbiome development. 

There are several limitations of our study. The sample size was limited to rigorously examine prenatal antibiotics as well as other factors such as excess infant weight at 1 year of age and ethnicity. Furthermore, although in general there was homogeneity in the type of peripartum antibiotics used, there was a lesser degree of prenatal antibiotics used, and a larger sample size would be needed to determine whether specific antibiotics and doses of antibiotics, broad versus narrow spectrum antibiotics or the specific timing in the prenatal period in which they were received, impacted the microbiome to a greater degree. Additionally, the dosing and duration of all of the prenatal antibiotics could not be confirmed. A further limitation is that meconium samples generally had a low abundance of bacteria and as with any low abundance microbiomes, there was a potential for contamination; our inclusion of both positive and negative controls serves to minimize this concern. Additionally, anthropometrics used were parent reported, which are known to have varying reliability. While measured anthropometrics were preferred, the accuracy of parent-reported anthropometrics was improved in this study by the removal of outliers, which has been previously validated in this cohort [28].

In conclusion, through examination of the meconium, which may provide an insight into intrauterine events and the very earliest pioneering gut microbes, it can be seen that prenatal and peripartum antibiotic use, in addition to CS delivery, were associated with differences in the structure and composition of the very early infant gut microbiome. Prospective longitudinal studies are needed to follow how early life antibiotic exposure may continue to affect microbiome development trajectories and association with diseases, such as obesity, thought to be related to aberrant gut microbiome development.

## Figures and Tables

**Figure 1 microorganisms-08-00179-f001:**
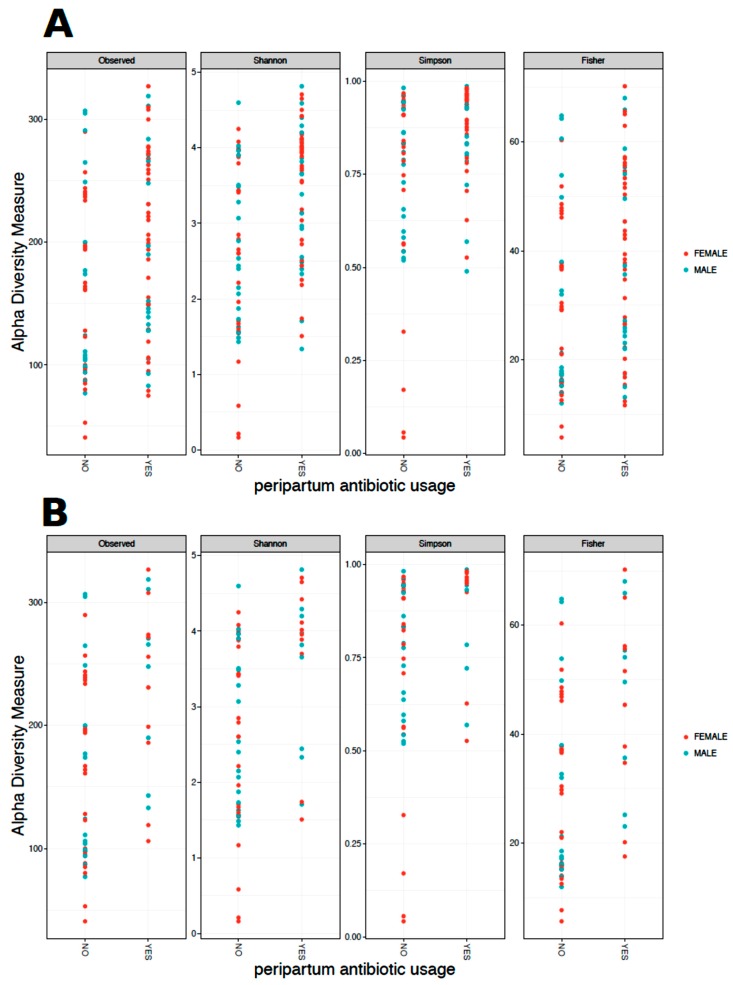
(**a**) Alpha diversity of 105 meconium samples stratified by peripartum antibiotic use (yes/no) and sex. Peripartum antibiotic use was associated with higher Shannon and Simpson diversity (*p* < 0.05), but a lower number of OTUs and Fisher diversity (*p* < 0.05). (**b**) Alpha diversity of meconium samples from infants born by vaginal delivery, stratified by peripartum antibiotic use (yes/no) and sex. Peripartum antibiotic use was associated with higher alpha diversity according to all four measures (*p* < 0.05, two sample Wilcoxon test).

**Figure 2 microorganisms-08-00179-f002:**
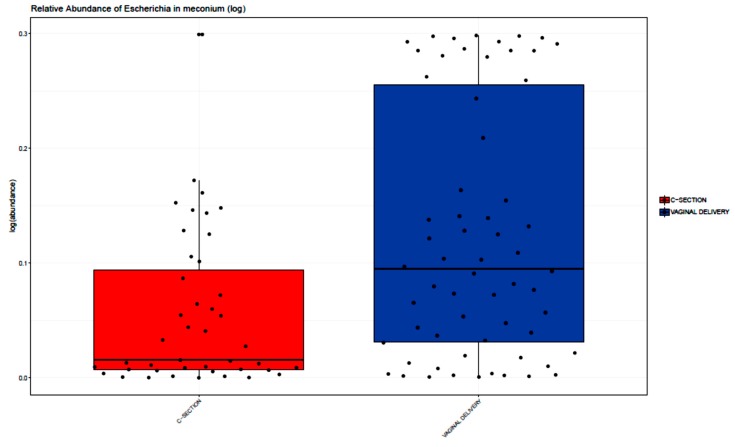
Relative abundance of *Escherichia* in meconium samples stratified by delivery mode. For each colored box: Horizontal bar = median, lower bar = first quartile (25th percentile), and upper bar = third quartile (75th percentile). *Escherichia* was higher in meconium samples from infants born by VD (38%) compared with CS (15%), FDR adjusted *p* < 0.001.

**Figure 3 microorganisms-08-00179-f003:**
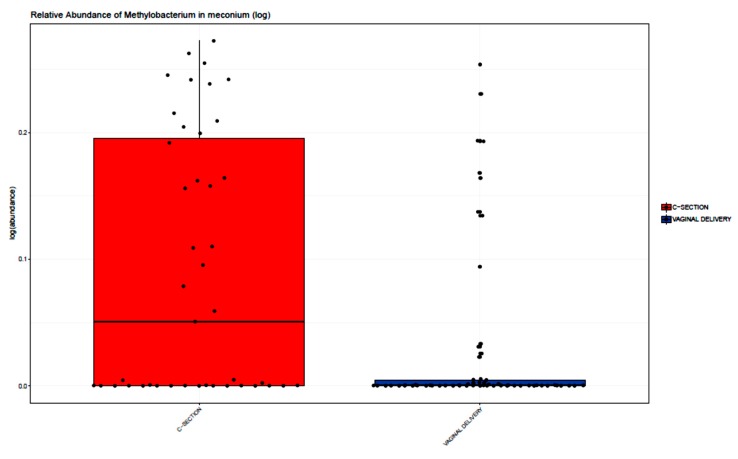
Relative abundance of *Methylobacterium* in meconium samples stratified by delivery mode. For each colored box: Horizontal bar = median, lower bar = first quartile (25th percentile), and upper bar = third quartile (75th percentile). *Methylobacterium* was higher in CS (27%) compared with VD (8%) adjusted *p* = 0.001.

**Figure 4 microorganisms-08-00179-f004:**
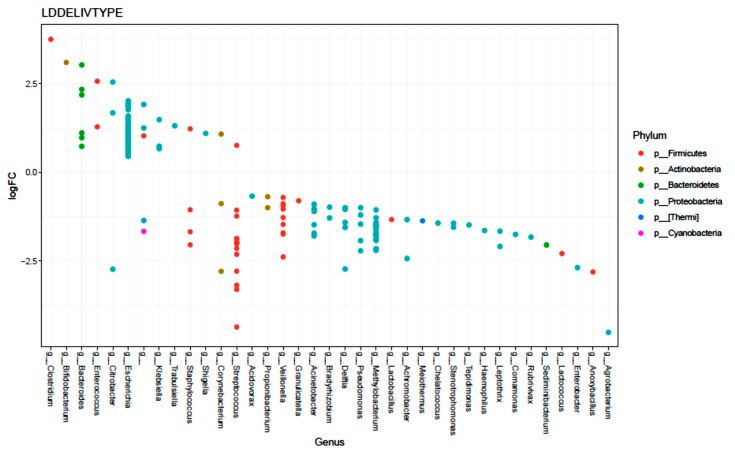
Differentially abundant OTUs (FDR < 0.05) according to delivery mode (negative log fold change (logFC) indicates OTU is enriched in meconium from CS infants, positive logFC indicates OTU is enriched in meconium from VD infants).

**Table 1 microorganisms-08-00179-t001:** Clinical features of infants by delivery mode.

Clinical Feature	Cesarean Section (n = 43)	Vaginal Delivery (n = 62)	*p* Value
Male	17 (40%)	28 (45%)	0.710
Scheduled Cesarean section	27 (63%)	N/A	N/A
Prenatal antibiotics	5 (12%)	12 (20%)	0.431
Peripartum antibiotics	43 (100%)	19 (31%)	**<0.001**
Not Hispanic ethnicity	20 (8 Hispanic, 15 unknown)	20 (13 Hispanic, 29 unknown)	0.538
Birth weight in grams (SD)	3283 (403)	3382 (357)	0.186
Infant weight for length percentile ≥85th% vs <85th % at 12 months (%) *	12/29 (41%) 14 unknown	17/34 (50%) 28 unknown	0.667
Infant weight for length percentile ≥95 vs <85th% at 12 months (%)	8/25 (32%) 14 unknown	10 /27 (37%) 28 unknown	0.928
Maternal pre-pregnancy body mass index in kg/m^2^ (SD)	26.8 (5.62)	26.7 (6.62)	0.949
Over the ACOG recommended weight gain pregnancy (%)	24 (56%)	22 (36%)	0.072

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
