# Peer review of "Prenatal and Peripartum Exposure to Antibiotics and Cesarean Section Delivery Are Associated with Differences in Diversity and Composition of the Infant Meconium Microbiome"

_microorganisms, 2020, doi:10.3390/microorganisms8020179_

Round 1

Reviewer 1 Report

This is an interesting manuscript describing the effect of antibiotics given to mothers prenatally and peri-natally on the microbiota content and diversity of the meconium.

The manuscript is well written and results are well presented. Even though the conclusions are weak, I find the study interesting and worthy of publication.   

As there are many confounding factor that may affect the results (most are mentioned in the discussion), in spite of the fact that a large number of children were included in the study; this makes the conclusions rather weak and sometimes difficult to follow. I suggest that the authors present the valid conclusions as a short summary.

As no association with obesity was indeed identified, I suggest that the introduction is revised accordingly.

Specific comments:

There is no need to use an abbreviation for Eppendorf tube. Why did the authors rely on parental reports regarding the anthropometric data at one year of age? Data may be inaccurate Discussion- lines 274-276 and lines 291-293 should be re-written

Author Response

Reviewer 1:

Comments and Suggestions for Authors

This is an interesting manuscript describing the effect of antibiotics given to mothers prenatally and peri-natally on the microbiota content and diversity of the meconium.

The manuscript is well written and results are well presented. Even though the conclusions are weak, I find the study interesting and worthy of publication.  

Thank you for your positive review of our manuscript and constructive comments. We hope our revisions have addressed these.

As there are many confounding factor that may affect the results (most are mentioned in the discussion), in spite of the fact that a large number of children were included in the study; this makes the conclusions rather weak and sometimes difficult to follow. I suggest that the authors present the valid conclusions as a short summary.

Thank you for this insightful comment. We agree. We have now added a short summary of the key valid conclusions at the start of the discussion.

As no association with obesity was indeed identified, I suggest that the introduction is revised accordingly.

Thank you. Given the only the modest findings of differential abundance of taxa with excess weight at 12 months of age, the emphasis on obesity has been decreased in the introduction and that specific paragraph shortened.

Specific comments:

There is no need to use an abbreviation for Eppendorf tube.

Thank you. The abbreviation has been removed.

Why did the authors rely on parental reports regarding the anthropometric data at one year of age? Data may be inaccurate

This is a great point. Unfortunately this data set is part of an ongoing longitudinal study in which only parentally reported data is collected. We appreciate that measured anthropometrics would be ideal. However in this cohort, parentally reported anthropometrics at 12 months with removal of outliers for reported lengths using the interquartile range, has previously been validated against anthropometrics measured in the pediatrician’s office (reference 28) – this has been clarified in the methods. In addition, the following has been added to the limitations section “An additional limitation is that the anthropometrics used are parent -reported, which are known to have varying reliability. While measured anthropometrics are preferred, the accuracy of parent-reported anthropometrics was improved in this study by removal of outliers, which has been previously validated in this cohort [28]. “

Discussion- lines 274-276 and lines 291-293 should be re-written

Thank you. These sentences have been rewritten for clarity.

Reviewer 2 Report

This manuscript reported the prenatal and peripartum exposure to antibiotics and Cesarean Section delivery alters the infant meconium microbiome. The work is well conceived and designed. However, the analysis and discussion in the present work should be improved before this study would be suitable for publication in Microorganisms.

1. In all figures, there was no statistical analysis.

2. In Lines 187-189, both Shannon and Simpson diversity increased. As we known, an increase in Simpson index indicates the lower microbiota community diversity, however, an increase in Shannon index suggest the higher microbiota community diversity. Thus, the author should explain what is the meaning of increases in Shannon and Simpson diversity.

3. The figure quality should be improved.

4. As we known, Escherichia is not a Good gut microbiota. CS versus vaginal delivery samples had lower Escherichia, the result should be discussed in the manuscript.

Author Response

Reviewer 2:

This manuscript reported the prenatal and peripartum exposure to antibiotics and Cesarean Section delivery alters the infant meconium microbiome. The work is well conceived and designed. However, the analysis and discussion in the present work should be improved before this study would be suitable for publication in Microorganisms.

Thank you for your positive review of our manuscript and constructive comments. We hope our revisions have addressed these.

In all figures, there was no statistical analysis.

Thank you for pointing this out. The statistical analysis in the main body text has now been clarified in the figure legends.

In Lines 187-189, both Shannon and Simpson diversity increased. As we known, an increase in Simpson index indicates the lower microbiota community diversity, however, an increase in Shannon index suggest the higher microbiota community diversity. Thus, the author should explain what is the meaning of increases in Shannon and Simpson diversity.

While all four alpha diversity measures indicate richness of the samples, the Shannon diversity and Simpson diversity also measures the evenness of a sample. Further, the Simpson index gives more weight to the more abundant species. Peripartum antibiotics was associated with increased Shannon and Simson diversity, but not Fisher or OTUs, suggesting that peripartum antibiotics may decrease some of the rarer taxa. This has now been explained in the alpha diversity results section of the manuscript.

The figure quality should be improved.

Thank you. We agree that the figures that are shown embedded in the current file are poor quality. However the high definition PDFs of the figures that were submitted along with the manuscript were of high quality and these should be the ones that are used at final publication. If there are any specific aspects of the high definition PDFs you would like us to change please let us know so and we would be happy to do this. In addition, we did notice that in supplementary figure 1, the figure legend was obscured and so this has been corrected and the new figure uploaded.  

As we known, Escherichia is not a Good gut microbiota. CS versus vaginal delivery samples had lower Escherichia, the result should be discussed in the manuscript.

Thank you. This is an excellent point and we agree this is true in the more developed microbiome. However, in neonates it is recognized that very early infant colonization is with facultative anaerobes, such as species from Escherichia and Enterococcus and, once the oxygen has been consumed, they are followed by strict anaerobes including Bifidobacterium, Bacteroides and Clostridium spp. Therefore the finding of Escherichia in the meconium of those born by vaginal delivery is not considered detrimental. This has now been clarified in the paragraph 3 of the discussion.